# GABA_A_ Receptor-Stabilizing Protein Ubqln1 Affects Hyperexcitability and Epileptogenesis after Traumatic Brain Injury and in a Model of In Vitro Epilepsy in Mice

**DOI:** 10.3390/ijms23073902

**Published:** 2022-03-31

**Authors:** Tabea Kürten, Natascha Ihbe, Timo Ueberbach, Ute Distler, Malte Sielaff, Stefan Tenzer, Thomas Mittmann

**Affiliations:** 1Institute of Physiology, University Medical Center of the Johannes Gutenberg University Mainz, Duesbergweg 6, 55128 Mainz, Germany; tkuerten@students.uni-mainz.de (T.K.); natascha_ihbe@hotmail.com (N.I.); tueberba@students.uni-mainz.de (T.U.); 2Institute for Immunology, University Medical Center of the Johannes Gutenberg University Mainz, Langenbeckstraße 1, 55131 Mainz, Germany; ute.distler@uni-mainz.de (U.D.); malte.sielaff@uni-mainz.de (M.S.); tenzer@uni-mainz.de (S.T.)

**Keywords:** traumatic brain injury, GABA_A_ receptors, in vitro epilepsy, E/I-balance, epileptogenesis, multi-electrode array, ubiquilin-1

## Abstract

Posttraumatic epilepsy (PTE) is a major public health concern and strongly contributes to human epilepsy cases worldwide. However, an effective treatment and prevention remains a matter of intense research. The present study provides new insights into the gamma aminobutyric acid A (GABA_A_)-stabilizing protein ubiquilin-1 (ubqln1) and its regulation in mouse models of traumatic brain injury (TBI) and in vitro epilepsy. We performed label-free quantification on isolated cortical GABAergic interneurons from GAD67-GFP mice that received unilateral TBI and discovered reduced expression of ubqln1 24 h post-TBI. To investigate the link between this regulation and the development of epileptiform activity, we further studied ubqln1 expression in hippocampal and cortical slices. Epileptiform events were evoked pharmacologically in acute brain slices by administration of picrotoxin (PTX, 50 μM) and kainic acid (KA, 500 nM) and recorded in the hippocampal CA1 subfield using Multi-electrode Arrays (MEA). Interestingly, quantitative Western blots revealed significant decreases in ubqln1 expression 1–7 h after seizure induction that could be restored by application of the non-selective monoamine oxidase inhibitor nialamide (NM, 10 μM). In picrotoxin-dependent dose–response relationships, NM administration alleviated the frequency and peak amplitude of seizure-like events (SLEs). These findings indicate a role of the monoamine transmitter systems and ubqln1 for cortical network activity during posttraumatic epileptogenesis.

## 1. Introduction

Traumatic brain injury (TBI) constitutes one of the leading causes of acquired epilepsy in young adults and is responsible for 10–20% of prevalent epilepsy cases [1,2]. The incidence of epilepsy is 30 times higher in patients affected by severe TBI in comparison to the general population [3]. Epileptogenesis after TBI is often characterized by enhanced seizure susceptibility [4,5] and the appearance of very high-frequency oscillations, so-called fast ripples [6]. The risk to develop posttraumatic epilepsy (PTE) is substantially elevated for more than ten years following TBI [7]. Consequently, this time window could be of great importance to start an effective anti-epileptogenic treatment. However, pharmacological attempts to prevent PTE emerge as highly difficult [3]. At present, the therapeutic options are restricted to alleviating the symptoms of acute seizures, treating comorbidities and preventing seizure recurrence [2].

Gamma aminobutyric acid (GABA)-mediated inhibition plays an essential role in regulating and controlling neuronal development and maintaining a physiological excitatory/inhibitory (E/I) balance by modulating excitatory synaptic transmission [8,9]. The reduced expression of GABA_A_ receptors and subunits [10] as well as an altered pre- and postsynaptic GABA_A_ and GABA_B_ receptor-mediated inhibitory neurotransmission [11,12] can promote an increased and enduring excitation and vulnerability of neuronal networks. TBI in mice has been reported to induce a loss of GABAergic interneurons associated with an enhanced glutamatergic signaling, resulting in a facilitation of neuronal excitation [13]. Therefore, such lesion-induced impairment of GABA-mediated transmission gives rise to the development of a hyperexcitable state in the brain [14,15]. Interestingly, the disturbances of focal TBI are not restricted to the vicinity of the lesion, but propagate to remote and uninjured brain areas as well, a phenomenon summarized under the term “diaschisis” [15,16]. In a former study performed in our lab by Le Prieult et al., neuronal hyperactivity and impaired GABAergic inhibition were observed in a mouse model of TBI in the intact contralateral hemisphere 24–48 h post-lesion [17]. GABA_A_ receptor-stabilizing protein ubiquilin-1 (ubqln1) interacts with the ubiquitin proteasome system (UPS) [18,19,20,21], which plays an essential role for maintaining proteostasis and the removal of misfolded accumulated proteins [22]. Ubqln1 accumulates at inhibitory synapses and functionally connects the ubiquitination machinery and polyubiquitinylated substrates with specific subunits of the proteasome [10,18,21]. At synaptic sites, the protein is co-localized with GABA_A_ receptor β2/3 subunits, extends their half-life and prevents early endoplasmic reticulum (ER)-related degradation [10,19,23,24]. Neurodegenerative diseases [25,26,27], epileptic disorders [10] and stroke-caused brain injuries [28] implicate the regulation by ubqln1 on the level of protein expression. The 63 kD protein, also referred to as plic-1, is downregulated in the progression of both epilepsies and brain injuries [10,28]. The number of receptors available for membrane insertion is enhanced through negative modulation of the proteasome mediated by ubqln1, as has been described for other ubiquitin-like proteins [18,23,24]. Recycled and newly synthetized GABA_A_ receptors are assembled within the ER followed by insertion into the neuronal plasma membrane, whereby ubqln1 is considered a regulatory determinant of membrane trafficking [10,24]. Therefore, the protein might play a key role in maintaining stable cell surface receptor numbers and enabling efficient inhibitory neurotransmission [23]. Without altering receptor internalization, ion channel function, anchoring and endocytic sorting, ubqln1 regulates the GABA_A_ receptor cell surface expression by stabilizing intracellular receptor pools within the secretory pathway [19,23,24].

The present study demonstrates that the expression of ubqln1 is strongly affected by TBI and epileptiform activity in mice. Zhang et al. reported decreased ubqln1 expression levels in human neocortical samples from TLE patients having at least 4 years of clinical history, and additionally in pilocarpine- and pentylenetetrazole (PTZ)-epilepsy models in rats [10]. In recent studies of stroke-induced brain injury and hypoxic ischemia, a significant decline in ubqln1 levels was observed 1–3 days post-injury [28]. Liu et al. were the first to deploy the non-selective monoamine oxidase (MAO) inhibitor nialamide (NM) in order to pharmacologically modulate ubqln1 expression and their treatment of striatal cell cultures with NM significantly upregulated ubqln1 expression [29]. Interestingly, the authors reported decreased ubqln1 levels following ischemia/reperfusion (I/R) injury in mice, which could be reversed by NM administration [29]. The MAO- and ubqln1-dependence of the observed effects were confirmed under knockout conditions [29].

In the present study, we quantified the impact of a TBI located in the motor-/somatosensory cortex in mice and in vitro epileptiform activity on cortical and hippocampal ublqn1 expression in Western blot experiments and its functional significance during electrophysiological multi-electrode array (MEA) recordings. We further discuss the recovering and neuroprotective effects of the non-selective MAO inhibitor NM.

## 2. Results

### 2.1. Altered Protein Expression of Ubqln1 in Cortical GABAergic Interneurons Isolated from the Contralateral Cortex 24 h Post-TBI

Previous data from our lab using the same unilateral TBI mouse model in the motor-/somatosensory cortex disclosed a transhemispheric diaschisis in terms of neuronal hyperexcitability exclusively contralateral to the lesion site, accompanied by an impairment of GABAergic inhibition 24 h post-TBI [17]. These results raised the question whether such unilateral TBI led to specific proteomic alterations in GABAergic interneurons in the undamaged contralateral hemisphere 24 h after induction of the injury. The present study aimed to identify target proteins in GABAergic interneurons potentially implicated in the disruption of neuronal networks and neurodegenerative sequelae that could serve as biomarkers or targets for novel therapies in TBI-affected patients in the future. Therefore, label-free quantification analysis was performed on isolated GABAergic interneurons. Cortices from the contralateral hemisphere of TBI-treated (*n* = 6) and sham-treated (*n* = 6) GAD67-GFP mice were isolated 24 h post-TBI (Figure 1A) to generate a single-cell suspension. GAD67-GFP positive interneurons and a GFP population of cells were isolated from the single-cell suspension using FACS, as described in [30]. We isolated a total of 56681 +/− 7278 GFP+ interneurons from the contralateral cortices of sham-treated animals and 46388 +/− 8901 GFP+ interneurons from TBI-treated animals (Figure 1B.i.). This resulted in the detection of 2030 +/− 360 proteins (sham, *n* = 9) in GFP+ lysates obtained from the sham group and 1794 +/− 241 proteins (TBI, *n* = 9) in lysates obtained from the TBI-treated group 24 h post-TBI by LC-MS and label-free quantification analysis (Figure 1B.ii.). The label-free quantification analysis revealed fifteen differentially expressed proteins in GFP+ GABAergic interneurons of the contralateral cortex 24 h post-TBI (Figure 1C; Appendix A). Twenty-two regulated proteins were identified in the GFP fraction of the single-cell suspensions obtained from sham- and TBI-treated mice 24 h after the unilateral damage (Appendix A). However, there was no overlap between the regulated proteins identified in the GFP+ interneurons and the GFP cells. This indicates that the regulated proteins display changes in their expression specifically in the GFP+ interneurons since all lysates were measured in the same experiment. The detected regulated proteins in the GFP+ interneurons of the contralateral cortex 24 h post-TBI (Figure 1C) were mainly associated with structural proteins. Interestingly, we could detect an increase in the protein expression of NFL (Neurofilament light polypeptide) in GFP+ interneurons. NFL has been described before as a biomarker in several neurological diseases, as it reflects axonal damage [31]. Other regulated proteins identified in GFP+ interneurons, such as Myh1 (Myosin-1), Myh7b (Myosin-7B) and TBA3 (Tubulin α-3 chain), are associated with tight junctions and could therefore indicate changes in network functions [32,33,34]. We considered the result of the TBI-induced decrease in the expression of ubqln1 in cortical GFP+ interneurons as particularly relevant in the context of cortical excitability. It has been shown that a regulation of ubqln1 is involved in the enhancement of seizure inhibition through the regulation of GABA_A_ receptors in epilepsy models in rats [10].

### 2.2. Regulation of Ubqln1 Expression in Cortex and Hippocampus 24 h Post-TBI

To validate our finding of altered ubqln1 protein expression in GFP+ interneurons (Figure 1), we performed quantitative Western blots of lysates obtained from whole cortices and hippocampi of TBI-treated animals 24 h after induction of the injury.

These Western blots confirmed that the reduced expression of ubqln1 in GFP+ interneurons of the contralateral cortex is not masked in lysates obtained from the whole, unsorted, contralateral cortex. The expression of ubqln1 was decreased in the contralateral cortex 24 h post-TBI using Kruskal-Wallis-test with Dunn’s multiple comparison (Figure 2A.i.; *n* = 8; *p* vs. sham: * 0.0401) as compared to cortical lysates from sham animals (*n* = 8). This effect was not observed in lysates of the contused ipsilateral cortex (*n* = 8; *p* vs. sham: ns; *p* vs. contralateral: * 0.0174). We also analyzed lysates of the hippocampi in the same animals 24 h post-TBI. Interestingly, the cortical lesion induced a decrease in ubqln1 expression in the hippocampus of the ipsilateral (Figure 2B.i.; *n* = 8; *p* vs. sham: * 0.0401) and contralateral (*n* = 8; *p* vs. sham: * 0.0233) hemisphere in comparison to lysates obtained from the hippocampi of sham-treated mice (*n* = 8).

### 2.3. Properties of Pharmacologically Induced Epileptiform Activity in Hippocampal Brain Slices during Multi-Electrode Array (MEA) Recordings In Vitro

In order to investigate the potential link between decreased ubqln1 expression in GABAergic interneurons early after TBI and altered excitability of the neuronal networks, which could possibly mediate the development of posttraumatic epilepsy, we evoked epileptiform activity in acute brain slices according to an established in vitro epilepsy model [35]. Administration of picrotoxin (PTX, 50 µM) and kainic acid (KA, 500 nM) to the bathing solution (mACSF) reliably evoked continuous epileptiform activity in acute brain slices (Figure 3). In our disinhibition model, seizure-like discharges occurred relatively synchronously over the whole recording period (Figure 3B). This experimental setup reliably induced repetitive seizure-like events (SLEs) in our brain slices, which enabled us to study alterations and modulation of epileptiform activity.

### 2.4. Epileptiform Activity Impaired Ubqln1 Expression in Hippocampal and Cortical Slices

Next, we focused on the effect of SLEs on the expression levels of ubqln1 in acute hippocampal and cortical slices. We quantified the protein expression at different time points following induction of epileptiform activity by bath-application of mACSF. Hippocampal and cortical tissue was examined separately for this purpose. Interestingly, the level of ubqln1 expression was diminished at all time points of incubation compared to control conditions in standard ACSF. Evaluation of the hippocampal protein levels (Figure 4A.ii.) disclosed a significant decline in ubqln1 expression after 1 h (*n* = 8; *p* vs. control: ** 0.0025) and 7 h (*n* = 8; *p* vs. control: * 0.0466), whereas cortical levels (Figure 4B.ii.) decreased after 1 h (*n* = 8; *p* vs. control: ** 0.0056), 5 h (*n* = 8; *p* vs. control: ** 0.0053) and 7 h (*n* = 8; *p* vs. control: ** 0.0035) of incubation in comparison to control levels in standard ACSF (*n* = 8). These findings strongly indicate that expression of the GABA_A_ receptor-stabilizing protein ubqln1 is impaired during epileptiform activity in a defined time window of up to 7 h after onset.

### 2.5. Nialamide, a Non-Selective Monoamine Oxidase Inhibitor, Rescued Expression of Ubqln1 in Hippocampal and Cortical Slices

It was demonstrated in previous studies that treatment with 10 µM of the MAO inhibitor nialamide (NM) can pharmacologically modulate and increase ubqln1 expression in cell culture experiments [29]. To test whether these findings apply to our epilepsy model, we examined protein levels following treatment with NM and under two experimental conditions: in control recordings and in our in vitro epilepsy model. For detection of a potential rescue, the epilepsy group (1 h of incubation in mACSF) lacking NM treatment was included in our Western blot analysis. Interestingly, NM treatment of slices under epileptic conditions led to a recovery of ubqln1 expression in the hippocampus (Figure 5A.ii.) and cortex (Figure 5B.ii.) reaching levels close to the untreated controls in standard ACSF. The effect was most prominent after 1 h of incubation and slightly subsided after 3 h and 5 h of incubation. In the hippocampus (Figure 5A.ii.), NM even increased the expression of ubqln1 after 1 h compared to epilepsy-treated slices without NM (*n* = 8; *p* vs. control + 1 h NM: * 0.0190). In the cortex (Figure 5B.ii.) a similar effect was observed between epilepsy conditions and the 1 h rescue group (*n* = 8; *p* vs. mACSF + 1 h NM: * 0.0303), and the control (*n* = 8; *p* vs. control: ** 0.0100) and the control with 1 h NM treatment (*n* = 8; *p* vs. control + 1 h NM: ** 0.0050). These data indicate that application of NM for 1 h is suitable to recover the expression of ubqln1 in both the hippocampus and the neocortex.

### 2.6. The MAO Inhibitor Nialamide Alleviated Epileptiform Discharges and the Peak Amplitudes in Slices during MEA Recordings

To evaluate any anti-epileptic effects of the MAO inhibitor, we applied NM during electrophysiological MEA recordings of acute CA1 hippocampal slices in our epilepsy model in vitro. By adding increasing concentrations of PTX (0–100 µM) to ACSF (containing 500 nM KA), we characterized the anti-epileptic effects of NM at different concentrations of the GABA receptor antagonist. This allowed us to generate dose-response curves to finally reach saturating levels of epileptiform activity. Since we observed increased ubqln1 expression after incubation with NM for 1 h in our Western blot experiments (Figure 5A.ii.), we also pre-incubated hippocampal slices with NM for 1 h before starting the MEA recordings. Then we analyzed the event frequency and peak amplitude (Figure 6B,C) under control conditions without the MAO-inhibitor NM (*n* = 9 slices/5 mice) and in the presence of NM (*n* = 15 slices/5 mice). Two-way ANOVA analysis revealed significantly lower levels of epileptic activity in the hippocampal slice preparations in the presence of the MAO inhibitor, with a reduced number of events (NM vs. no NM ** *p* = 0.0061) and smaller peak amplitudes (NM vs. no NM *** *p* = 0.0004). Furthermore, our findings suggest that in the presence of low concentrations of PTX (10 μM), the MAO inhibitor almost fully prevented epileptic activity, most likely by raising the threshold for the generation of epilepsy in vitro (Appendix A, Mann–Whitney test: NM vs. no NM * *p* = 0.0119).

In summary, our results from the Western blot and the electrophysiological MEA-experiments provide evidence that the MAO inhibitor NM rescues the expression of the GABA_A_ receptor-stabilizing protein ubqln1 during pharmacologically induced in vitro epilepsy in the hippocampus and neocortex. It has been demonstrated by Le Prieult et al. in our lab that unilateral TBI leads to early cortical hyperexcitability and an impaired GABAergic inhibition [17]. In the present study, the early application of NM impaired the observed epileptic activity in the hippocampal network in our in vitro slice model, possibly through its stabilizing actions on GABA_A_ receptors, as revealed by the recovered expression levels of ubqln1.

## 3. Discussion

It is still not fully understood how an altered protein expression after TBI mediates dysfunctions of the brain such as posttraumatic epilepsy or other TBI-related neurodegenerative sequelae. Although transhemispheric diaschisis could play a major role in the functional reorganization and recovery after focal brain injury, little is known about the cellular mechanisms underlying this phenomenon [15,17]. Therefore, the present study used a proteomic approach to identify target proteins regulated in GABAergic interneurons of the contralateral cortex as early as 24 h post-TBI. The analysis disclosed a group of proteins, which were mainly linked to structural and molecular functions. In addition, we could identify protein neurofilament light chains (NFL), known as a robust biomarker for TBI and other neurological disorders [31,36,37,38,39]. Incidentally, our dataset also uncovered new proteins, which could mediate the strength of cortical GABAergic inhibition after TBI, thereby contributing to the development of neurological disorders such as posttraumatic epilepsy (PTE) [7]. Especially our observation of decreased expression of ubqln1 identified in contralateral GABAergic interneurons raised our attention, since it might be an important player in the recovery process of TBI patients in the early phase at 24 h after the injury. So far, one study reported ubqln1 to be involved in the modification of GABA_A_ receptor cell surface expression and in the progression of epilepsy. Here the reduced expression and binding of ubqln1 to GABA_A_ receptors correlated with hyperexcitability and a lowering of the seizure threshold, as recorded by increased seizure severity and shortened seizure onset [10]. Interestingly, our lab also observed cortical hyperexcitability associated with altered GABAergic inhibition in a mouse model of TBI in the early phase of 24 h to 48 h following the induction of the injury [17]. We hypothesized that our observation of an early impaired GABAergic inhibition after TBI is strongly linked to altered ubqln1 expression, as our present data disclosed regulation of ubqln1 exclusively in cortical GABAergic interneurons 24 h post-TBI.

Regulation of ublqn1 early after TBI was also detected in Western blot lysates in the cortex, demonstrating that altered expression was present in the whole cortex and not exclusively limited to GABAergic interneurons. Therefore, we cannot exclude that the protein is also downregulated in other cell types. Due to its known stabilizing effects on GABA_A_ receptors [10,19,23,24] we decided to further investigate the functions of ubqln1 during cortical network hyperexcitability in an in vitro mouse model of cortical disinhibition-induced epilepsy that would mimic the TBI-related impairment of GABAergic signaling [9,40]. The combination of the two substances PTX and KA evoked reproducible and highly repetitive SLEs in our MEA recordings and was implemented successfully before by Ridler et al. [35]. We expected that hippocampal and cortical ubqln1 levels would be severely affected by epileptiform activity in this in vitro model in an acute time window of up to 7 h. Although we measured ubqln1 expression in the whole hippocampus, it might be possible that one subfield is more affected than the others. In TLE animal models, neurodegeneration and a severe loss of GABA_A_ receptor subunits have been reported in CA1 and CA3 subfields of the hippocampus [41]. In recent CCI studies, altered GABAergic inhibitory currents and their recovery after TBI were measured, especially in the CA1 subfield [42,43]. In animal epilepsy models, fast ripples, epileptiform activity and postsynaptic GABA_A_ receptor function have been extensively studied in CA1 [44,45]. Hence, we conducted our electrophysiological recordings primarily in the CA1 region. The observed decline in ubqln1 expression in both models, in vivo TBI and in vitro epilepsy, might directly lead to an enduring propensity towards epileptiform activity, as we consider it plausible that GABA_A_ receptor subunits might destabilize and lose their integrity in the neuronal plasma membrane without ubqln1. Reduced ubqln1 levels might be linked to earlier ER-associated degradation of GABA_A_ receptors and their subunit isoforms [10,23]. Subsequently, physiological synaptic inhibition could be attenuated and compromised by emerging glutamatergic excitatory network oscillations [13]. To achieve a potential recovery of ubqln1 expression in our in vitro epilepsy model, we included the MAO inhibitor NM in our study, which would eventually reverse the observed reduction in ubqln1 levels. Western blot quantification confirmed our hypothesis that ubqln1 levels could be fully recovered through NM administration. As suggested by recent studies, we assumed that modulating ubqln1 expression would help to regain physiological GABA_A_ receptor stabilization and hypothesized that an upregulation of ubqln1 expression might alter the properties of evoked epileptiform activity in the hippocampal CA1 region and schaffer collaterals itself. With respect to the mean number of events and the mean peak amplitude, the course of the PTX-dependent concentration curves differed significantly in our electrophysiological recordings (Figure 6). Strikingly, the difference was also evident at high PTX concentrations of 50–100 µM that theoretically block nearly all GABA_A_ receptors. Therefore, we conclude that the implied effects of NM are not necessarily based on strengthened GABA_A_ receptor-mediated inhibition or restricted to an increased ubqln1 expression. Yet, it seems likely that the monoamine transmitter systems might be involved in this attenuation of hyperexcitability. The non-selective inhibitor of both MAO-A and MAO-B nialamide (NM) of the hydrazine class interacts irreversibly with both isoforms of the monoamine oxidases [29,46]. The non-selective inhibition indirectly increases the amount of available extracellular monoamine transmitters and biogenic amines. The oxidative deamination through MAOs generates reactive oxygen species such as hydrogen peroxide (H_2_O_2_) or noxious byproducts such as ammonia (NH_3_), causing oxidative stress and damage to mitochondria [29,47,48], both implicated in epileptogenesis [49]. Therefore, the protection from oxidative stress is an additional positive side effect of NM and other MAOIs, contributing to the neuroprotective potential of the drug [29,50]. Like other non-selective MAO inhibitors, NM was originally devised for atypical and treatment-resistant forms of depressive disorders [29,51]. In spite of being effective however, it has been withdrawn from the market due to adverse effects such as hypertension [29,51]. However, long-term research data and randomized double-blind clinical trials emphasize the effectiveness of MAOIs and the requirement to integrate their safe usage more into education and clinical expertise [52]. Although MAO inhibitors and monoamine transmitters have been discussed controversially in terms of treating epileptic disorders, studies indicate that the activation of 5HT_1A_-serotonine and D_2_-dopamine receptors display anticonvulsive effects [53,54,55]. This clarifies the great potential of medication for epilepsy patients that operates and interacts with the monoamine transmitter system. In the future it might become possible to treat further drug-resistant forms of epilepsy and posttraumatic epilepsies that remain particularly difficult to treat. Prospectively, ubqln1 might be an effective target for novel therapies in PTE-patients. In future studies, its role in functional reorganization and recovery following TBI could be further assessed. Moreover, spreading depolarization (SD) is a neurological phenomenon associated with both seizures and TBI [56,57]. According to clinical studies in TBI-patients, SD might be a prognostic factor for less favorable outcomes [56,58]. Therefore, pre-clinical investigations of SD in TBI- and epilepsy models could gain valuable data for developing treatment strategies for patients suffering from PTE.

*Conclusions:* The present study gives new insights into the role of ubqln1, particularly its regulation and modulation in CCI-treated mice and in vitro epilepsy. Ubqln1 represents a target with neuroprotective potential, while its altered expression might play a key role in posttraumatic epileptogenesis. Our findings go in line with the reported decrease in protein expression in the course of epileptic activity and various neurological diseases. By treating acute slices with the MAO inhibitor NM, we were able to restore physiological ubqln1 levels. Furthermore, our electrophysiological data present evidence for anti-epileptic effects of NM in hippocampal slices.

## 4. Materials and Methods

### 4.1. Animals and Ethical Approval

All experiments were conducted in accordance with European standards for the use of animals in research and the institutional regulations of the Johannes Gutenberg University, Mainz, Germany. Experiments were permitted by the local Animal Ethics Committee of the Landesuntersuchungsanstalt Rheinland-Pfalz, Germany, permission numbers 23 177-07/G20-1-112 and 23 177-07/G15-1-039. C57BL/6n wild-type mice (*n* = 53) at the age of 17–24 days and GAD67-GFP mice (*n* = 12) at the age of 19–22 days of either sex were used in this study (*n_total_* = 65). They had free access to food and water and were housed in a standard 12 h light/dark cycle at a constant room temperature of 23 ± 2 °C. All efforts were made to minimize their suffering.

### 4.2. Controlled Cortical Impact

GAD67-GFP (*n* = 12) and C57BL6/N mice (*n* = 12) received a controlled cortical impact (CCI) to the motor and sensorimotor cortex as described before by our lab [17]. Originally, the GAD67-GFP mice were generated by Tamamaki et al. [59]. Mice were initially anesthetized at the age of 19–22 days postnatal with 4 Vol% isoflurane (Abbvie, Wiesbaden, Germany) and the anesthesia was obtained by application of 1–2 Vol% isoflurane over a face mask. The head was fixated in a stereotactic frame, the fur of the upper head was shaved, the scalp disinfected with ethanol and treated with lidocaine-containing cream (Elma salve, Aspen GmbH, Germany). A median incision of 1 cm length was induced to expose the cranial bone. Next, a cranial window of 4 mm^2^ starting from bregma and extending caudal over the right hemisphere was drilled and opened without damaging the dura matter. To remove contamination from the cranial window, the surface was cleaned using Dulbecco’s Phosphate-buffered saline (DPBS, Lonza, Basel, Switzerland). The trauma was induced with a depth of 0.8 mm, a speed of 6 m/s, an impact duration of 200 ms and an impactor tip with 1.5 mm diameter using a CCI-impactor (Impact One^TM^; Leica Mikrosysteme, Wetzlar, Germany). The cranial window was closed with cyanoacrylate-based tissue adhesive (3M Vetbond, 3M Animal Care Products, St. Paul, MN, USA) using the skull bone of the window. Lastly, the wound was closed with polypropylene sutures (Ethicon, Somerville, NJ, USA), then the anesthesia was terminated, the animal was placed back in the cage and allowed to recover under infrared light. Sham-operated littermates were treated similarly but did not receive a cranial window or impact. The sham-treated group served as a control group.

### 4.3. Electrophysiology In Vitro

#### 4.3.1. Preparation of Brain Slices

The mice were deeply anesthetized with 4% isoflurane. Nociception was tested to assure deep anesthesia before decapitation. Brains were immediately removed and transferred to ice-cold oxygenated standard artificial cerebrospinal fluid (ACSF) containing in mM 125 NaCl, 26 NaHCO_3_, 3 KCl, 1.3 MgSO_4_·7H_2_O, 2.5 CaCl_2_·H_2_O, 1.25 NaH_2_PO_4_, 13 D-glucose. Acute horizontal sections from both brain hemispheres with a thickness of 400 µm were prepared using a vibratome (Leica VT-1200S, Leica Mikrosysteme, Wetzlar, Germany). Throughout the procedure, slices were consecutively transferred onto a nylon net and incubated in ACSF equilibrated with 95% O_2_ and 5% CO_2_ (pH = 7.4) at room temperature.

#### 4.3.2. In Vitro Epilepsy Model

We implemented an epilepsy model, which would mimic the disinhibition and impaired GABAergic transmission as observed in our TBI-injury model [9,40,60]. Hence, we chose the allosteric modulator picrotoxin (PTX) that prevents ion flow through the chloride channels of GABA_A_ receptors [61]. The in vitro epilepsy model was intended for extracellular multi-electrode array (MEA) recordings in hippocampal slices and for hippocampal and cortical Western blot lysates. We used the established in vitro epilepsy model by Ridler et al. [35] by adding a combination of picrotoxin (PTX, cat. No. 1128, Tocris Bioscience, Bristol, UK) and kainic acid (KA, cat. No. 0222, Tocris Bioscience, Bristol, UK) to our standard ACSF. The ACSF containing 50 μM of PTX and 500 nM of KA was defined here as modified ACSF (mACSF). As hippocampal regions are the most vulnerable to abnormal network oscillations [10,62,63], horizontal brain sections were optimal for extracellular detection of neuronal activity and local field potentials.

#### 4.3.3. Multi-Electrode Array Recordings

Extracellular multi-unit recordings in spatially defined hippocampal regions were conducted by utilization of square MEA chips (60MEA200/30iR-Ti-gr, Multichannel Systems, Reutlingen, Germany). They were equipped with 59 planar TiN-electrodes and one internal reference; the inter-electrode distance was 200 µm and the electrode diameters reached 30 µm. One MEA-chip was utilized for multiple recordings of various experimental conditions. Prior to data acquisition, the acute brain sections were transferred onto the MEA-chip and left to incubate for 30 min. A platin grid was placed on top to improve electrode contact and secure the position of the brain slice. Temperature was maintained constantly at 32 °C using a temperature controller (TC02, Multichannel systems, Reutlingen, Germany) and water bath for the applied ACSF. Constant perfusion with oxygenated ACSF was ensured via tube pump system (Minipuls 3, Gilson, Middleton, WI, USA) and perfusion cannulas. All extracellularly acquired signals occurred spontaneously and were monitored over recording periods of 10 min, respectively. The sampling rate was set to 50 kHz and raw signals were recorded with the MEA2100-acquisition system. No filters were applied.

Epileptiform activity was evoked pharmacologically via bath-application of PTX and KA in mACSF as described above. Dose–response relationships were investigated by administration of increasing PTX concentrations (0, 10, 25, 50, 100 µM) in the presence of KA (500 nM) and optionally nialamide (NM, 252999, Sigma-Aldrich, St. Louis, MO, USA). The increasing PTX doses were washed in for approximately 15 min prior to data acquisition. The effect of the non-selective MAO inhibitor NM on epileptiform activity was examined in comparison to the control condition devoid of NM treatment. Slices were pre-incubated in ACSF containing 10 µM of NM for at least 1 h according to our Western blot results after 1 h. NM washout was performed in the presence of mACSF to assure that our observations were NM-dependent and partially reversible in the amplitude of seizure-like events. Using the Multichannel Analyzer, a single electrode in the hippocampal CA1 region or schaffer collaterals was selected per slice and exported for offline analyses.

Epileptic events were detected via threshold-based Clampfit analysis (pCLAMP 11.1, Molecular Devices, Sunnyvale, CA, USA) to determine the mean number of discharges and mean peak amplitude. The threshold was set relatively low to 30–50% of the max. amplitude at a PTX concentration of 100 µM. Per slice and concentration curve, the same threshold was maintained respectively, while the baseline was constantly set to 0. The detection threshold ranged from min. 0.015 mV to max. 0.2 mV.

### 4.4. Western Blots

Quantification of ubiquilin-1 (ubqln1) expression levels was accomplished with sodium dodecyl sulfate polyacrylamide gel electrophoresis (SDS-PAGE). To evaluate protein levels following the induction of epileptiform activity, acute 400 µm-thick horizontal slices were prepared as described and incubated in mACSF. At defined time points of incubation (1, 3, 5, 7 h), several slices were extracted for the preparation of hippocampal and cortical lysates. After 1 h of incubation in the mentioned in vitro epilepsy model (mACSF), NM was added to the incubation chamber in order to ascertain an NM-mediated recovery of ubqln1 expression levels. The effect of NM was determined after 1, 3 and 5 h of incubation and further analyzed under non-epileptic control conditions. Lysates were prepared from 8 mice of each experimental condition. Sterile scalpel blades were utilized to carefully separate hippocampal and cortical areas. Liquid nitrogen immediately cooled down the separated tissue in Lo Bind reaction tubes (Eppendorf SE, Hamburg, Germany), which were then stored at −80 °C. Subsequently, the tissue was homogenized in lysis buffer containing N-PER™ Neuronal Protein Extraction Reagent (87792, Thermo Fisher Scientific, Waltham, MA, USA), Halt™ Protease and Phosphatase Inhibitor Cocktail and 0.5 M EDTA solution (78440, Thermo Fisher Scientific, Waltham, MA, USA, ratio 100:1:1).

For Western blots 24 h post-TBI, CCI-treated animals were anesthetized with 4 Vol% isoflurane and decapitated. The brain was removed, transferred into ice-cold DPBS and the cortices as well as the hippocampi isolated under sterile conditions. The tissue was flash frozen in liquid nitrogen. Whole cortices were lysed using 1.5 mL and hippocampi using 0.5 mL N-PER Neuronal Protein extraction reagent (Thermo Fisher Scientific, Waltham, MA, USA) containing the Halt Protease and Phosphatase Inhibition cocktail (1:100; Thermo Fisher Scientific, Waltham, MA, USA).

After centrifugation of the epilepsy and TBI probes at 4° C (13000 rpm for 15 min), supernatants were extracted. Bicinchoninic acid assays (Pierce™ BCA Protein Assay Kit, 23225, Thermo Fisher Scientific, Waltham, MA, USA) and photometric measurement determined the total protein concentration of each sample. Aliquots were completed by adding dithiothreitol (DTT, 43815, Sigma-Aldrich, St. Louis, MO, USA) and lithium dodecyl sulfate sample buffer (4×) (LDS, Nu^®^ PAGE, NP0007, Thermo Fisher Scientific, Waltham, MA, USA) and adjusting the exact amount of lysate and lysis buffer. For electrophoretic separation by molecular weight, equal amounts of protein were applied (20 µg per lane) and migrated towards the anode through 10% separating gel before being transferred to polyvinylidene diflouride (PVDF) membranes. Afterwards, the membranes were blocked in 4% non-fat dry milk (diluted in Tris-buffered saline + Tween 20 (TBS-T)) for 45 min at room temperature. For immunostaining, the primary anti-ubiquilin polyclonal antibody (1:1000, PA1-759, Thermo Fisher Scientific, Waltham, MA, USA) incubated overnight at 4 °C. The housekeeping protein glyceraldehyde 3-phosphatase dehydrogenase (GAPDH, 36 kD) was quantified for normalization; its primary antibody (1:1000, A303-878A, Bethyl Laboratories, Montgomery, TX, USA) was incubated for 1 h at room temperature. The membranes were washed with TBS-T for 30 min with 10 min changes in between. Secondary antibodies were horseradish-peroxidase (HRP) conjugated (HRP anti-rabbit antibody: 1:10,000, Jackson ImmunoResearch, West Grove, PA, USA and HRP anti-goat antibody: 1:10,000, Dianova, Hamburg, Germany) and incubated for at least 30 min followed by enhanced chemiluminescent (ECL) detection (Immobilon Western HRP Substrate, WBKLS0500, Millipore, Burlington, MA, USA). Images were created with the chemidoc (ChemiDoc XRS+, Bio-Rad Laboratories, Hercules, CA, USA) and ImageLab 2.0 software (Bio-Rad Laboratories, Hercules, CA, USA). Image Studio Lite 5.2 Software (LI-COR Biosciences, Lincoln, NE, USA) was utilized for densitometric quantification.

### 4.5. Fluorescence-Activated Cell Sorting (FACS)

Fluorescence-activated cell sorting (FACS) was used to isolate GFP+-labeled GABAergic interneurons of cortices from CCI- and sham-treated GAD67-GFP mice (*n* = 12) as detailed in (30). Therefore, the mice were anesthetized with 4 Vol% isoflurane and decapitated 24 h post-TBI (traumatic brain injury). Only the contralateral undamaged cortex was isolated and transferred to ice-cold DPBS. The cortices were dissociated using the adult brain dissociation kit protocol (Order #130-107-677; Milentyi Biotec, Bergisch Gladbach, Germany) and the gentleMACS Octo Dissociator (Milentyi Biotec, Bergisch Gladbach, Germany). The obtained single-cell suspension was filtered using MACS SmartStrainer (70 µm; Milentyi Biotec, Bergisch Gladbach, Germany). Prior to the cell sorting, cells were pelleted with a centrifugation step at 4 °C and 300× *g* for 10 min (Centrifuge 5424R, Eppendorf, Hamburg, Germany) and resuspended in 200 µL DPBS. In order to separate a GAD67-GFP+ and a GFP cell population, the BD FACSAria III Cell Sorter (BD, Franklin Lakes, NJ, USA) operated by the BD FACSDiva 8.0.2. Software (BD, Franklin Lakes, NJ, USA) was used with an 85 μm nozzle. The gating for the cell sorting was used as established in (30). DAPI (4′,6-Diamidino-2-phenylindol) was added to the cell suspension to exclude dead cells, and Side Scatter Area (SSC-A) and Side Scatter Width (SSC-W) were used to exclude doublets. Cells were sorted into 2 mL Protein LoBind tubes (Eppendorf AG, Hamburg, Germany) containing 50 μL DPBS, pelleted at 2000 rpm for 10 min at 4 °C and flash frozen in liquid nitrogen prior the protein extraction for label-free quantification analysis.

### 4.6. Liquid Chromatography Mass Spectrometry Analysis and Label-Free Quantification Analysis

Proteins of sorted cells were extracted and digested as described before in [30]. Here, the optimized single-pot solid-phase-enhanced sample preparation (SP3) protocol was used [64]. In brief, cell pellets were resuspended in urea-based lysis buffer, sonicated in a Bioruptor (Diagenode, Liège, Belgium) for 15 min at 4 °C, and Dithiothreitol (DTT) was added for reduction, along with iodoacetamide (IAA) to alkalize the samples. Next, 40 μg carboxylate-modified paramagnetic beads (Sera-Mag Speed Beads, Thermo Fisher Scientific, Waltham, MA, USA) were reconstituted in water and added to the lysates; 70% (*v*/*v*) Acetonitrile (ACN) was added to promote protein binding to the beads and incubated for 18 min. Beads were immobilized on a magnetic rack and washed twice with 70% (*v*/*v*) ethanol (EtOH), once with 100% ACN, and resuspended in 50 mM ammonium bicarbonate. Trypsin was added for overnight digestion at a ratio of 1:25 (*w*/*w*, trypsin/protein) at 37 °C. Lastly, 95% (*v*/*v*) ACN was added, the supernatant discarded, and the beads resuspended in 2% (*v*/*v*) dimethyl sulfoxide (DMSO) to elute the peptides from the beads. For liquid chromatography mass spectrometry (LC-MS) analysis, 1% (*v*/*v*) formic acid (FA) was added to the peptides.

The tryptic peptides were analyzed as described before in [65,66]. Here, a NanoAQUITY UPLC system (Waters Corporation, Milford, MA, USA) was coupled online to a Synapt G2-S high-definition mass spectrometer (Waters Corporation, Milford, MA, USA) via a NanoLockSpray dual electrospray ion source (Waters Corporation). Peptides were loaded directly onto an HSS-T3 C18 1.8 μm, 75 μm × 250 mm reversed-phase analytical column (Waters Corporation) running a gradient from 5 to 40% (*v*/*v*) mobile phase B (0.1% (*v*/*v*) FA and 3% (*v*/*v*) DMSO in ACN) at a flow rate of 300 nL/min over 90 min. The mobile phase A consisted of 0.1% (*v*/*v*) FA and 3% (*v*/*v*) DMSO in water. LC-MS analysis of eluting peptides was performed as described before by Distler et al. [65,66] by ion-mobility separation (IMS)-enhanced data-independent acquisition (DIA) in UDMSE mode.

The ProteinLynx Global Server (PLGS, ver.3.0.2, Waters Corporation) was used for raw data processing. Therefore, data were compared to a custom compiled mouse proteome database (UniProtKB mouse reference proteome release 2017) that also contained a list of common contaminants. For the database search: (1) Trypsin was reported as an enzyme for digestion, (2) up to two missed cleavages per peptide were allowed, and (3) peptides had to have a length of at least six amino acids to be considered. Further, carbamidomethyl cysteine was set as fixed, and methionine oxidation as variable modification. To assess the false discovery rate (FDR) for peptide and protein identification the target-decoy strategy was used. Therefore, a reverse database was searched in PLGS and the FDR was set to 0.01.

For postprocessing and label-free quantification analysis, ISOQuant ver.1.8 was used as described in detail by Distler et al. [65,66]. Proteins were only reported if they had been identified by at least two peptides with a minimum length of six amino acids, a minimum PLGS score of 5.5, no missed cleavages and an FDR below 0.01. Absolute in-sample amounts were calculated for each protein using TOP3 quantification as described in [67]. Significantly regulated proteins (*p* < 0.05) were identified by applying a two-tailed *t*-test and post hoc multiple-hypothesis testing using Bonferroni correction. Therefore, only proteins that were discovered in five or more samples were considered.

Heatmaps of identified significant regulated proteins were generated using Perseus 1.6.10.43 (open-source-software, Max Planck Institute of Biochemistry, Computational Systems Biochemistry, Martinsried, Germany).

### 4.7. Statistical Evaluation

For statistical analysis, Graph Pad Prism 8 software (GraphPad Software, San Diego, CA, USA) was used. All results are represented as Mean ± Standard Error of Mean (SEM). For non-parametric distribution, Kruskal–Wallis with Dunn’s multiple comparison and pairwise Mann–Whitney tests were performed. Two-Way ANOVA was deployed for statistical comparison of dose–response relationships and curve progressions. For liquid chromatography mass spectrometry analysis and label-free quantification analysis, two-tailed *t*-test and post hoc multiple-hypothesis testing with Bonferroni correction were performed. Asterisks indicate significantly different values in the figures (*: *p* < 0.05; **: *p* < 0.01; ***: *p* < 0.001).

### 4.8. Limitations

All experiments were performed as an ex vivo–in vitro study. Pharmacological evocation of epileptiform activity in hippocampal regions was conducted according to an established in vitro epilepsy model [35]. Despite its high functionality and reproducibility in extracellular recordings, the applicability of a pharmacologically induced epilepsy model in mice in vitro to the human form of an epileptic disorder is limited. PTE is a multifactorial disorder, as brain injuries cause manifold disturbances in the human brain. Individual and neuropathological predispositions, such as cortical dysplasia leading to a higher risk of PTE, have to be considered here [68,69]. In future studies, a non-disinhibition epilepsy model would allow a differentiation between a general impact of epileptiform activity on ubqln1 expression and the effect of PTX-induced disinhibition. NM was originally developed for human medication but was discontinued many years ago due to adverse drug reactions. Therefore, a deployment of selective MAO inhibitors or other serotonin and dopamine-interacting drugs could provide valuable information on the role of the specific monoamine transmitter in epileptogenesis. Additionally, targeting ubqln1 more specifically with another drug or by using ubqln1-knockout animals could reinforce our hypotheses. In traumatic brain injury, the reported downregulation of ublqn1 in cortical GFP+ interneurons 24 h contralateral of the lesion site might not be restricted to this particular cell type since our additional Western blots showed an overall decrease in whole cortices and hippocampi. Therefore, we cannot exclude that ubqln1 expression is regulated in other cell types as well. Additional in vivo animal studies would be useful to monitor the long-term course of ubqln1 expression in a chronic model of posttraumatic epilepsy.

## Figures and Tables

**Figure 1 ijms-23-03902-f001:**
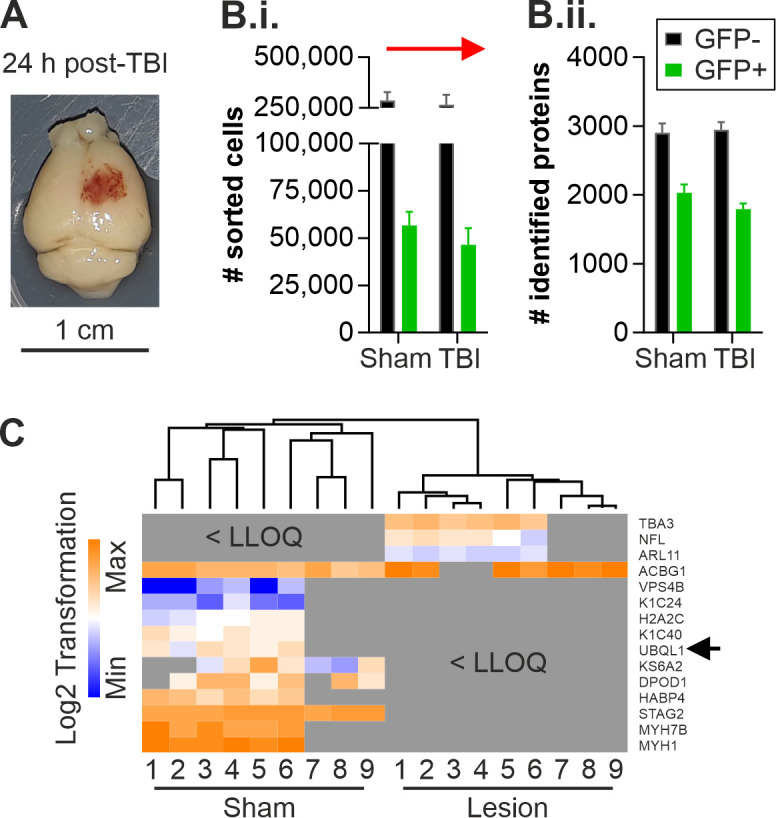
Label-free quantification of GAD67-GFP interneurons of the contralateral cortical cortex 24 h post-TBI. (**A**) Representative TBI-treated and PFA-fixed brain of a GAD67-GFP mouse 24 h post-TBI. Coagulated blood is left in the lesion site after perfusion due to the disruption of the blood–brain barrier. (**B.i.**) GAD67-GFP+ interneurons and GFP− cells were isolated from the contralateral cortex and separated using FACS. The number of isolated and sorted cells did not differ between sham- and TBI-treated animals. (**B.ii.**) The number of identified proteins also did not differ between the sham- and TBI-treated groups. (**C**) Relative protein abundances obtained using LCMS with label-free quantification analysis are displayed in a log2 transformed heat map (blue = low abundance, orange = high abundance, grey = below lower level of quantification, LLOQ). Fifteen regulated proteins were identified in GAD67-GFP interneurons 24 h post-TBI, mostly associated with structural functions. The protein expression of ubqln1 was decreased in GFP+ interneurons of the contralateral cortex 24 h post-TBI (see arrowhead).

**Figure 2 ijms-23-03902-f002:**
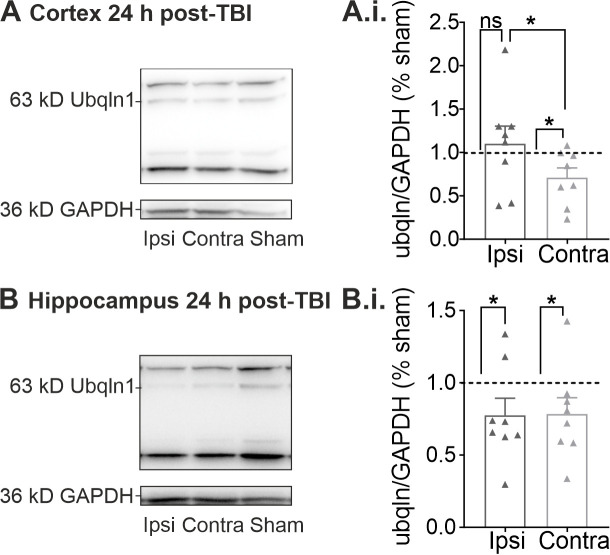
Quantitative Western Blots of lysates obtained 24 h post-TBI. (**A**) Representative molecular weight bands for ubqln1 (63 kD) and the housekeeper protein GAPDH (36 kD) of lysates obtained from the whole cortex ipsilateral (ipsi), contralateral (contra) and from sham mice 24 h post-TBI. (**A.i.**) The ubqln1 signal per GAPDH signal was normalized to the intensities detected in lysates obtained from sham mice. Note that the expression of ubqln1 was reduced only in the contralateral cortex 24 h post-TBI. (**B**) Representative weight bands for ubqln1 and GAPDH obtained from hippocampus lysates 24 h post-TBI. (**B.i.**) The ubqln1 per-GAPDH expression was lowered in both hippocampal lysates of the contralateral and the ipsilateral hemisphere 24 h post-TBI. The data are represented as mean ± SEM and significant values were detected using a Kruskal–Wallis test with Dunn’s multiple comparison, indicated here as * *p* < 0.05.

**Figure 3 ijms-23-03902-f003:**
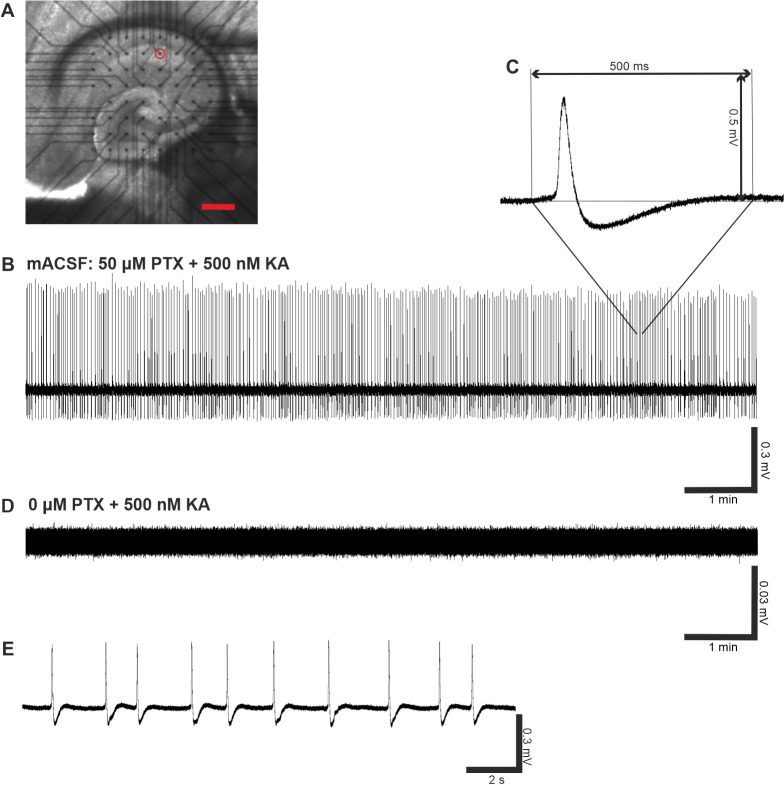
MEA recordings of epileptiform activity in hippocampal brain slices in vitro. (**A**) Position of the 400 µm-thick horizontal brain slice on top of one MEA chip. Scale bar = 400 μm. Black dots indicate the location of 60 recording electrodes incl. one internal reference electrode. The diameter of each electrode was 30 μm and the inter-electrode distance reached 200 µm. For data acquisition and analysis, one channel (red circle) in the CA1 region was selected. (**B**) Pharmacologically evoked epileptiform activity in vitro using modified artificial cerebrospinal fluid (mACSF) containing 50 μM picrotoxin and 500 nM kainic acid. Representative 10 min voltage trace revealed spontaneous activity in the hippocampal CA1 region with continuous occurrence of seizure-like events over the whole recording period. (**C**) Example of a single seizure-like event at higher temporal resolution and (**D**) representative voltage trace in absence of PTX, but in presence of 500 nM KA. Note the higher magnification of the voltage signal for better display of noise resolution. (**E**) Characteristic 18 s segment depicting 10 SLEs, taken out of the representative trace above in (**B**).

**Figure 4 ijms-23-03902-f004:**
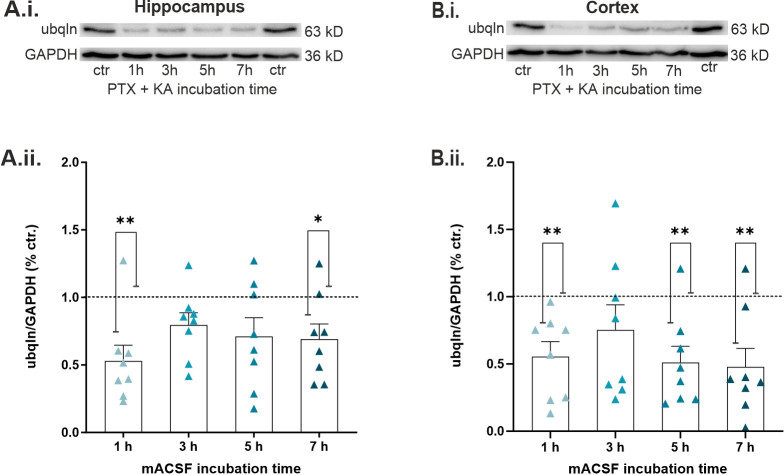
Expression levels of Ubiquilin-1 (ubqln1) in our in vitro epilepsy model in hippocampal and cortical slices incubated in mACSF containing PTX and KA. (**A**,**B**) Typical Western blot weight bands of hippocampal (**A.i.**) and cortical (**B.i.**) slices acquired 1-7 h after induction of epileptiform activity. Expression of ubqln1 (63 kD) was quantified and normalized to the housekeeping protein GAPDH (36 kD). (**A.ii.**) Ubqln1 levels in the hippocampus (*n* = 8 animals) and cortex (**B.ii.**) (*n* = 8 animals) are displayed in scatter plots, normalized to the protein expression level of the control in standard ACSF. Protein levels of the control condition are represented by the dashed line and normalized to 1. The data in the scatter plots are represented as mean ± SEM, indicated by bars and errors bars. For statistical analysis, Kruskal–Wallis test was performed to disclose significantly different values, indicated as ** *p* < 0.01, * *p* < 0.05. Note the significantly decreased ubqln1 expression after 1 h (** *p* = 0.0025) and 7 h (* *p*= 0.0466) in hippocampus (**A.ii.**). Analysis of the cortical protein expression revealed a significant decline after 1 h (** *p* = 0.0056), 5 h (** *p*= 0.0053), and 7 h (** *p* = 0.0035) of incubation (**B.ii.**).

**Figure 5 ijms-23-03902-f005:**
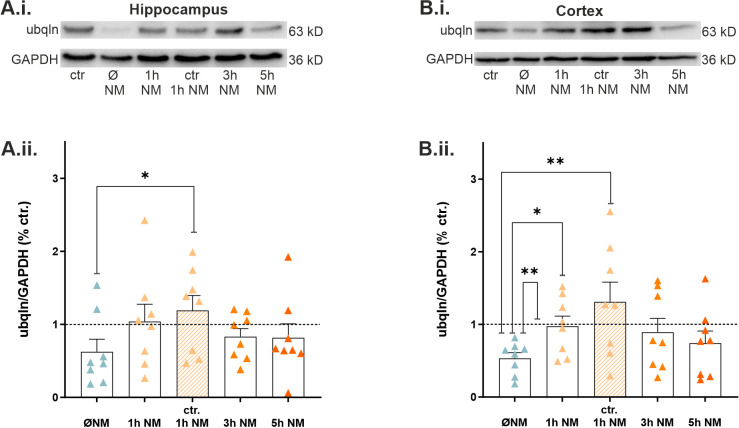
Expression of hippocampal and cortical levels of ubqln1 following treatment with nialamide (NM) during epilepsy in vitro. (**A.i.**) Representative Western blot weight bands of lysates from hippocampal and cortical (**B.i.**) slices. (**A.ii.**) Scatter plots of hippocampal and cortical (**B.ii.**) ubqln1 (63 kD) expression at various time points of incubation in mACSF or standard ACSF and subsequent treatment with 10 µM NM. The scatter plots summarize data from *n* = 8 mice, and the data were normalized to the signal of the housekeeping protein GAPDH. The dashed line represents the control group that was kept in standard ACSF and was set to 1. Bar plots indicate mean values ± SEM. Asterisks represent significantly different values received from Kruskal–Wallis tests; they are indicated as * *p* < 0.05, ** *p* < 0.01. Note the recovery of ubqln1 expression in the hippocampus during epilepsy in the presence of the MAO inhibitor. We observed even an increase in ubqln1 expression between data from epilepsy conditions in the absence of NM and data from control tissue bathed in standard ACSF, but in the presence of NM (* *p* = 0.0190). In the cortex, increased expression was disclosed between the epilepsy group in the absence of NM and the control group (** *p* = 0.0100), as well as the epilepsy group + 1 h NM (* *p* = 0.0303) and the control + NM (** *p* = 0.0050), respectively.

**Figure 6 ijms-23-03902-f006:**
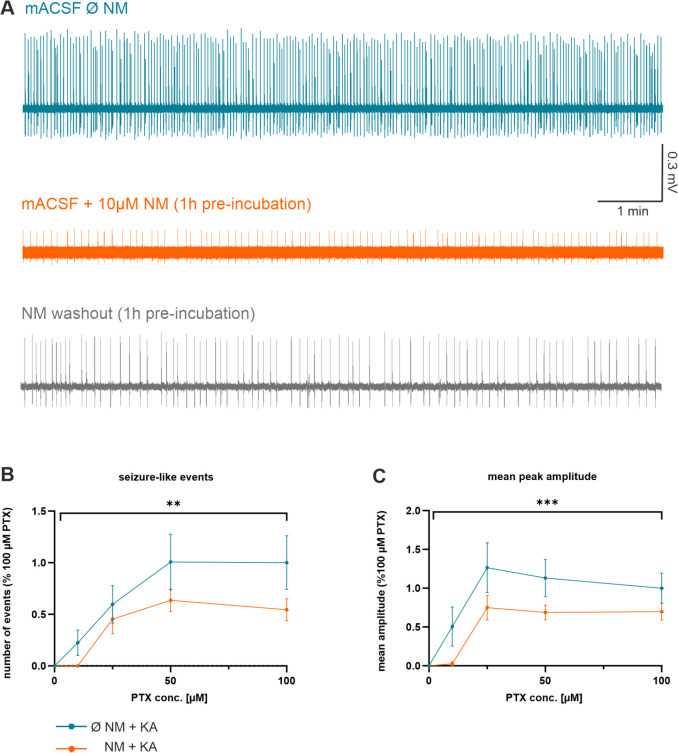
The MAO inhibitor Nialamide reduced the number of events and the mean peak amplitudes in extracellular MEA recordings in slices during epilepsy conditions in vitro. (**A**) Representative voltage traces of epileptiform activity in the hippocampal CA1 region induced by 50 µM PTX and 500 nM KA. The top trace was recorded in mACSF; the middle trace was acquired in mACSF after incubation for 1 h in standard ACSF containing NM (10 µM); the bottom trace was recorded after incubation for 1 h in ACSF containing NM (10 µM) and subsequent washout of NM in mACSF. (**B**) Summary diagram showing the effects of NM on the number of epileptic events recorded at increasing concentrations of bath-applied PTX and of 500 nM KA in the absence (*n* = 9 slices/5 mice) or presence of NM (*n* = 15 slices/5 mice). (**C**) Summary diagram showing the effects of NM on the peak amplitude of epileptic events recorded at increasing concentrations of bath-applied PTX and of 500 nM KA in absence (*n* = 9 slices/5 mice) or presence of NM (*n* = 15 slices/5 mice). All data points in B and C are represented as relative values normalized to mean numbers recorded in PTX (100 µM). Two-way ANOVA analysis revealed impaired epileptic activity in the presence of the MAO inhibitor concerning the number of epileptic events (NM vs. no NM ** *p* = 0.0061) and mean peak amplitude (NM vs. no NM *** *p* = 0.0004).

## Data Availability

The mass spectrometry proteomics data presented in this study have been deposited to the ProteomeXchange Consortium via the jPOST partner repository (https://doi.org/10.1093/nar/gkw1080; https://pubmed.ncbi.nlm.nih.gov/27899654/; accessed on 31 January 2022) with the dataset identifiers PXD031368 (ProteomeXchange) and JPST001466 (jPOST). Western blot data presented in this study are available as a ZIP-file with the Appendix A. Electrophysiological raw data from this study are available on request from the corresponding author.

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
