# Peer review of "GABAA Receptor-Stabilizing Protein Ubqln1 Affects Hyperexcitability and Epileptogenesis after Traumatic Brain Injury and in a Model of In Vitro Epilepsy in Mice"

_ijms, 2022, doi:10.3390/ijms23073902_

Round 1

Reviewer 1 Report

The authors present an interesting study regarding insights into GABAa-receptor-stabilization protein ubiquilin-1 in GAD67-GFP mice undergoing unilateral TBI. The study is rigorously designed and authors should be applauded for the work. 

Figure 1: the GFP+ interneurons had decreased expression of ubqln1 at 24 hours. Data is sound. Please provide further explanation why the contralateral cortex was investigated. Is this a suspected coup/contra-coup phenomenon. 

Figure 2: interesting findings. With rodent studies, difficult to isolate specific regions of hippocampus involved. Please expand in discussion how different regions may be preferentially altered such as CA1 vs. CA2. 

Figure 3: data is valuable and rigorous. Any plans to investigate spreading cortical depression? 

Figure 4: data is interesting. Please discuss rationale for choosing the specific convulsive agents. 

Figure 5: data is convincing. Any plans to investigate a more specific agent to target ubqln1 more specifically? 

Figure 6: future study it would be beneficial to look at spreading cortical depression as well. Please add small section to discussion. 

This paper is well written, presents a novel concept, and data rigorous. It would benefit from expansion of discussion into the topics mentioned above as well as strategies to expand further in quest to get to 1st in human trials. Overall I recommend acceptance with minor revisions. 

Reviewer 2 Report

Major issues

#1. By this research, the authors lead the result of “MAO-inhibitor NM rescues expression of the GABA-receptor stabilizing protein ubqln1 during pharmacologically induced in vitro”. However, “These findings indicate a role of the monoamine transmitter systems in the recovery of the excitatory-inhibitory (E/I) balance in posttraumatic epileptogenesis epilepsy in hippocampus and neocortex.

”  was not fully proven by this study. Please modify the expression.

#2. The authors sincerely stated the limitation of the study. Just consider saying that brain injury related epilepsy is multi factorial. This is also important to apply this idea to human individuals.

10.1016/j.yebeh.2020.107352

10.1089/neu.2015.4220

Minor issues

#1. In Abstract, there is a typo of expression.

#2. Typo of development 

#3. E/I balance can be understood by some specialists. Please indicate full words at the first use.

#4.UB-tagged also needs full words.

#5. ER-need full words.

#6. Section 4.3.2 MEA alone but section 4.3.3 with full words.

#7. Even though GABA is well understood. However, this also needs full words, I think.  

Please see all abbreviations. The author used them inconsistently.  
